# Decoding Quantitative Traits in Yaks: Genomic Insights for Improved Breeding Strategies

**DOI:** 10.3390/cimb47050350

**Published:** 2025-05-12

**Authors:** Yujiao Fu, Yuanyuan Yu, Xinjia Yan, Daoliang Lan, Jiabo Wang

**Affiliations:** Key Laboratory of Qinghai-Tibetan Plateau Animal Genetic Resource Reservation and Utilization, Ministry of Education and Sichuan Province, Southwest Minzu University, Chengdu 610041, China; 240710002010@stu.swun.edu.cn (Y.F.); 240710002014@stu.swun.edu.cn (Y.Y.); yan@swun.edu.cn (X.Y.)

**Keywords:** yak, genomic prediction, molecular-associated selection, breeding

## Abstract

The yak (*Bos grunniens*), the only large domesticated species endemic to the Qinghai–Tibet Plateau, is a vital resource for local livelihoods and regional economic sustainability. However, yak breeding faces significant challenges, including limited understanding of the genetic architecture underlying quantitative traits, inadequate advanced breeding strategies, and the sterility of hybrid offspring from yak–cattle crosses. These constraints have hindered genetic progress in key production traits. To address these issues, integrating modern genomic tools into yak breeding programs is imperative. This review explores the application and potential of molecular marker-assisted selection (MAS) and genomic prediction (GP) in yak genetic improvement. We systematically evaluate critical components of genomic breeding pipelines, including: (1) phenotypic trait assessment, (2) sample collection strategies, (3) reference population design, (4) high-throughput genotyping (via genome sequencing and SNP arrays), (5) predictive model development, and (6) heritability estimation. By synthesizing current advances and methodologies, this work aims to provide a framework for leveraging genomic technologies to enhance breeding efficiency, preserve genetic diversity, and accelerate genetic gains in yak populations.

## 1. Introduction

The yak (*Bos grunniens*) is the only large domestic animal in the Qinghai–Tibet Plateau, which is valuable economically and ecologically. The yak industry plays an essential role in the economy and culture of the Qinghai–Tibet Plateau. Over 17.6 million yaks are distributed worldwide, with the majority inhabiting the ~2.5 million km^2^ plateau regions of Central Asia, primarily centered on the Qinghai–Tibetan Plateau and its adjacent highlands [1]. In such high-altitude regions, where alternative sources of animal protein are scarce, yaks serve as vital providers of essential proteins, including high-quality meat protein, nutrient-rich milk protein, and collagen-derived proteins, thereby fulfilling an irreplaceable ecological and economic niche. For herders, yaks are not only a source of meat, milk, hides, and labor but also a cultural and religious symbol, directly supporting the livelihoods and nutritional security of plateau populations [2,3]. Yak husbandry is primarily pastoral, involving seasonal grazing, minimal infrastructure, and a strong dependence on natural grasslands.

Modern breeding on the plateau faces multiple real-world constraints. Previous evidence suggests that the domestic yak populations were domesticated during the Holocene and had two deeply divergent phylogenetic groups [4]. The primary breeding goals in domestic yak populations include body size, environmental adaptability, and load-bearing capacity. Currently, phenotypic selection remains the predominant method used by herders for breeding decisions [5]. Breeding decisions are largely based on herders’ experience and observable traits, with limited use of pedigree records or genetic evaluation. This results in slow genetic progress and the potential for inbreeding. Economically, herders often lack access to advanced genetic tools due to limited resources, and the industry remains semi-industrialized. Logistically, sample collection is hindered by the remoteness of grazing areas and the difficulty of sample preservation and transportation. Ethically, balancing the conservation of rare local breeds with commercial breeding goals remains challenging. These issues emphasize the urgency of developing practical, scalable, and inclusive breeding strategies. Current efforts focus on improving traits such as meat quality, milk yield, and stress resistance [6]. However, the genetic improvement of yak quantitative traits has been relatively slow due to the limited understanding of their underlying genetic mechanisms, the lack of advanced breeding selection methods, and the issue of male sterility in hybrid offspring from yak and cattle crosses. Interspecies hybridization between yaks and domestic cattle improves production traits, with F1 and F2 hybrids showing enhanced milk and meat yields and strong adaptability to high-altitude, harsh environments. However, nearly all F1 hybrid males are sterile, despite normal libido and secondary sex traits, due to testicular failure in producing viable sperm [7]. This infertility often persists until the seventh–eighth backcross generation, accompanied by reduced hybrid vigor. Although yaks and cattle share the same chromosome number (2n = 60), structural differences and chromosomal rearrangements may disrupt meiosis [8]. Recent molecular studies have identified candidate genes and dysregulated pathways involved in spermatogenesis, offering insights into the genetic and epigenetic mechanisms behind hybrid male sterility. These findings are critical for improving reproductive efficiency and guiding future yak breeding strategies.

Genomic selection or genomic prediction (GS or GP) has become a key research tool in modern animal and plant breeding, as well as in the treatment and prevention of major human diseases. It allows the estimation of breeding values and prediction of future phenotypes at an early age by GP modeling, significantly improving the selective accuracy and greatly enhancing the efficiency of selection in animal and plant breeding. Introduced in 2008, VanRaden first replaced the pedigree relationship matrix with a molecular relationship matrix to predict breeding values for milk yield in dairy cattle, marking the introduction of the Genome Best Linear Unbiased Prediction (gBLUP) method in livestock and crop breeding [9].

The initial domestication of yaks and the gradual breeding process were primarily based on phenotypic selection, where individuals with larger body sizes and stronger production performance were selected as breeding stock to produce offspring. With the continuous development of biology, especially molecular biology and sequencing technologies, key molecular markers or genes that influence phenotypes can be used to estimate marker-estimated breeding values (mEBV). This has led to the widespread application of marker-assisted selection (MAS) in genetic evaluation and breeding selection of livestock and poultry [10]. Currently, MAS has achieved success in improving traits related to disease resistance, milk protein composition, and meat quality. Several important candidate genes have been reported to be associated with major yak production traits, such as EPAS1 and HIF1A (related to high-altitude adaptation), MSTN (affecting muscle growth), and CSN2 (regulating milk protein content) [11,12,13]. In 2014, the key gene FOXL2 that controls the horned or hornless phenotype in ruminants was identified [14], and selection based on this gene type can effectively control the presence or absence of horns in offspring. Subsequently, Yan Ping’s laboratory initiated the breeding of hornless yaks and obtained official breed certification in 2019 [15]. This represents one of the most notable cases of MAS-based breeding in yaks and has encouraged researchers to apply MAS for breeding evaluation and selection.

Although GP holds great potential in yak breeding, it still faces numerous challenges, including difficulties in measuring production traits and specific phenotypes, the low application rate of gene chips in specialized breeding, the lack of genome-wide prediction tools optimized for the yak’s genetic background, and the complexity of the genetic mechanisms underlying multi-trait inheritance. Combining MAS with GS could offer a more comprehensive breeding strategy. On the one hand, MAS can be used for targeted selection based on known functional genes, while on the other hand, GS can estimate the Genomic Estimated Breeding Values (GEBV) for complex traits based on whole-genome SNP data, thereby improving breeding efficiency. This narrative review explores the application and development of these technologies in yak breeding, aiming to provide references for enhancing the economic benefits of yaks, protecting their genetic diversity, and promoting genetic progress in yak populations (Figure 1). This review first starts with the introduction of the main target traits in yak breeding. Then, the role and method of tissue sample collection and the construction of a reference population in the breeding plan will be presented. Finally, the results of the published and relevant literature between 2000 and 2025 on different applications of breeding approaches, such as MAS, GS, and the estimation of GEBV, will be critically discussed.

## 2. Phenotypic Measurements

Even individuals with identical genotypes may have different phenotypes under varying environmental conditions. Influential environmental factors include diet, climate, stress, disease, and so on [16]. Phenotypic measurement plays a crucial role in breeding [17], genetic research [18], and disease studies.

Considering the challenges and unique aspects of collecting yak trait data, phenotypic measurements can be divided into normal phenotypes and specific phenotypes.

### 2.1. Normal Traits

Normal phenotypic measurements provide fundamental data. Body size and weight are the basic indicators of a yak’s production performance. Measuring traits such as body weight (BW), chest girth (CG), body length (BL), and wither height (WH) at different growth stages can evaluate the growth rate and feed conversion efficiency.

#### 2.1.1. Body Size Measurement

With the development of artificial intelligence, efficient systems for yak body size measurement have been successively developed. Yan et al. used image analysis and regression models to accurately estimate yak body weight based on body measurements [19]. Wang et al. proposed a contactless method combining body point extraction algorithms and depth imaging to measure yak body size and weight [20]. Zhang et al. developed a machine vision-based method for yak weight estimation on edge devices and established a comprehensive display system for yak estimation based on user interactions with the edge devices [21].

Modern technologies for an automated phenotypic measurement of yaks improve farming standardization, labor efficiency, animal welfare, and data accuracy. However, several limitations persist. The accuracy of image and depth-sensing systems can be affected by variable outdoor conditions. Model generalizability is often limited to specific yak populations, and high equipment costs hinder adoption in remote areas. Moreover, measurement depends on the animal’s posture and cooperation, while the absence of standardized evaluation frameworks limits consistency and comparability across methods. Therefore, while AI-based systems hold great promise, further efforts are needed to enhance robustness, reduce costs, and establish unified evaluation standards to ensure broader and more reliable applications in yak farming.

#### 2.1.2. Production Performance Measurement

Production performance measurement mainly includes evaluating various productivity factors, such as meat production, reproductive rate, and carcass weight. It can improve farming efficiency, reduce costs, and enhance economic returns by monitoring and optimizing these traits.

Reproduction is a critical aspect of yak farming, determining herd expansion and long-term profitability. Generally, yaks typically reach sexual maturity at 3–4 years and can reproduce normally by 6–7 years, producing 4–5 calves over their lifetime [22]. As seasonal breeders, their mating and conception are influenced by temperate seasons [23]. Stillbirth and abortion rates range from 5% to 10%. Reproductive traits such as age at first calving, calving interval, conception rate, and number of calves per lifetime are collected through on-farm records or periodic surveys, but the reliability of these data is often compromised by inconsistencies in recording practices and limited sampling frequency. Hormone assays [24] and ultrasound diagnosis [25] are sometimes used to confirm reproductive status in field studies and improve diagnostic precision.

### 2.2. Specific Traits

While normal phenotype measurements provide basic data, considering specific phenotypes is crucial for improving livestock productivity and profitability. Common specific phenotype assessments include multiple ribs and high-altitude adaptation.

#### 2.2.1. Multiple Ribs

The Jinchuan yaks have unique anatomical features. It has been reported that 52% of Jinchuan yaks have 15 thoracic vertebrae and 15 corresponding pairs of ribs, with one additional thoracic vertebra and one extra pair of ribs [26]. Yaks are categorized into normal and variant types based on the thoracic and lumbar vertebrae numbers. Normal yaks have 14 thoracic vertebrae and five lumbar vertebrae (T14L5), while the variants include T15L5, T15L4, and T14L6, representing 3%, 8%, and 37% of the population [27]. Variations in thoracic and lumbar vertebrae may influence body length, particularly thoracic expansion. Additional thoracic vertebrae can enlarge the chest cavity, enhancing oxygen capacity, while extra lumbar vertebrae improve load-carrying ability during transportation.

Multiple ribs traits are assessed through external palpation and X-ray scanning. In T15L5 variants, the last rib’s morphology resembles that of T14L5 yaks, necessitating the use of external palpation alongside other indicators like body conformation and weight to avoid misjudgments that could lead to the loss of valuable genetic resources. Advanced X-ray methods, such as CT scans, have been used to measure rib and skeletal structures in pigs, analyzing their relationship with carcass and growth traits [28]. The beneficial multiple ribs mutation has significant potential for practical applications.

#### 2.2.2. High-Altitude Adaptation

Yaks have evolved unique morphological, physiological, and biochemical traits to adapt to the harsh environments of the Qinghai–Tibetan Plateau, which features low oxygen, high altitude, and extreme cold [29,30,31]. Morphological adaptation is a physical change that occurs over multiple generations in animals, enhancing their adaptability to a given environment. Compared to cattle in low-altitude regions, yaks have relatively larger lungs and hearts. Their compact bodies, dense outer fur, and lack of functional sweat glands improve cold resistance [32]. In addition to low oxygen levels, freezing temperatures and scarce food further exacerbate challenges. Yak tongue morphology allows them to consume a diverse range of plants, improving feed digestibility [33]. Additionally, their rumens can process large amounts of low-quality forage through prolonged fermentation, extracting maximum nutrients during scarce periods [34].

The evaluation of high-altitude adaptability in yaks involves a combination of morphological measurements, hematological analyses, and molecular assessments. Morphologically, chest girth, body condition, and hair density are commonly recorded as indicators of cold resistance and oxygen storage capacity. Hematological parameters such as the hemoglobin concentration, red blood cell count, and hematocrit levels provide quantitative insights into the yak’s oxygen-carrying efficiency under hypoxic conditions. At the molecular level, the expression or variation of the genes involved in hypoxia response (*EPAS1*, *HIF1A*, and *EGLN1*) is detected through genotyping or transcriptomic analysis, offering reliable genetic markers for evaluating altitude adaptation [35,36].

## 3. Sample Collection Method and Reference Population Construction

### 3.1. Sample Collection Method

Considering the unique nature of yak sample collection, such as the pastoral management and inherent aggressiveness of the animals, blood collection using physical restraint is considered the safest and most effective method. Typically, venous blood collection is performed, including jugular vein, tail vein, and femoral vein sampling. In addition to blood, other biological samples such as hair follicles, feces, and mucosal swabs can also be collected for genetic or health assessments. Hair follicles are typically obtained by plucking, providing valuable DNA samples. Fecal samples are collected non-invasively and are useful for studying gut microbiota or parasite burdens. Mucosal swabs, often taken from the oral or nasal cavity, are useful for detecting pathogens or assessing immune responses. Muscle tissue from yaks is typically collected during slaughter for performance assessment and is thus mainly utilized within reference populations due to limited availability. These diverse sampling strategies offer essential resources for comprehensive physiological, genetic, and health-related studies in yaks.

### 3.2. Reference Population Construction

The reference population is a crucial component in GS, used to evaluate the overall genetic parameters, and serves as the foundation for genomic breeding work. However, due to the complex structure of yak populations under free mating, difficulties in determining relatedness, and the presence of multiple local breeds, the composition and size of the reference population need to be determined based on the breeding goals (intra-breed selection or inter-breed crossbreeding improvement). The reference population must possess sufficient genetic diversity and representativeness to ensure the reliability and broad applicability of the research results. In practical production, the reference population should include as many descendants, full siblings, and half-siblings of the breeding bulls as possible. Through scientific design and implementation, the reference population can provide solid data support for genomic research, breeding improvements, and the enhancement of production traits in yaks.

## 4. Genomic Sequencing and Gene Chip Technology

### 4.1. Development of the Yak Reference Genome

The reference genome is the foundation and core of species genome research. A high-quality, comprehensive, and accurate reference genome is necessary for genomic selection studies, enabling researchers to identify genetic variations related to traits at the genomic level, thereby advancing precision breeding and genetic improvement. Yak reference genome studies primarily focus on two subspecies: the domestic yak (*Bos grunniens*) and the wild yak (*Bos mutus*) [37].

In 2012, researchers published the first yak reference genome, marking a significant milestone in the genomic research of this species [29]. This study laid the foundation for subsequent genomic analyses, allowing for an in-depth study of yak genomes, including the exploration of genetic diversity, evolution, and adaptive traits. Currently, for the yak reference genome, the Northwest Plateau Institute of Biology of the Chinese Academy of Sciences, in collaboration with other institutions, has utilized second- and third-generation sequencing technologies [38], along with Hi-C interaction mapping technology to assist with genome assembly, creating high-quality chromosome-level reference genomes for both wild and domestic yaks.

With the completion of the high-quality yak reference genome assembly, GWAS (Genome-Wide Association Studies) has also been widely applied in yak breeding. These studies mainly focus on yak growth and development traits, as well as reproductive, meat quality, and slaughter traits [39,40,41].

### 4.2. Transcriptome Sequencing and Single-Cell Sequencing in Yak Research

In yak research, RNA sequencing (RNA-seq) and single-cell sequencing technologies have unveiled gene expression patterns that regulate specific traits under different physiological conditions, providing crucial data for elucidating the roles of key genes within regulatory networks (Figure 2) [42]. By comparing the transcriptome data of lung tissues from yaks living at high and low altitudes, researchers identified differentially expressed genes (DEGs) associated with hypoxia adaptation, such as *CD36* and *PRKCSH*. These genes are involved in the regulation of blood cell proliferation, arachidonic acid metabolism, and ovarian steroidogenesis pathways, contributing to yak survival in low-oxygen environments [43]. Ge et al. performed mRNA, lncRNA, and miRNA sequencing of yak lung tissues collected from individuals at altitudes of 3400 m, 4200 m, and 5000 m, identifying altitude-associated DEGs such as *PDIA4*, *BAX*, and *CAPN1*. These genes are involved in immune response and cell cycle regulation, which may enhance yak adaptation to hypoxic conditions [44]. In 2022, Gao et al. performed transcriptomic analysis to construct a yak lung cell atlas, identifying a distinct subtype of endothelial cells. They identified 127 genes carrying structural variants (SVs) with a high fixation index (FST) that exhibited differential expression between yak and cattle lung tissues. SVs within the promoter regions of these genes may be involved in gene expression regulation under hypoxic conditions. Additionally, several hypoxia-regulated genes carrying SVs, such as *ARNT*, *GATA1*, *MAFG*, *KLF5*, and *HOXB5*, were identified [45].

To adapt to hypoxic conditions of high altitudes, yak cardiac tissue exhibits distinct gene expression and protein regulation mechanisms. Studies have shown that hypoxia-inducible factor 1α (*HIF-1α*) is widely expressed in yak heart, liver, and lung tissues, with the highest expression observed in heart tissue [46]. As a key regulatory factor, *HIF-1α* maintains cardiac function by modulating angiogenesis, erythropoiesis, and glycolysis. Wang et al. identified six key genes related to hypoxia adaptation in the yak heart through transcriptome sequencing: *MAPKAPK3*, *PXN*, *NFATC2*, *ATP7A*, *DIAPH1*, and *F2R*. These genes are likely involved in regulating myocardial energy metabolism and signal transduction, facilitating yak adaptation to hypoxic high-altitude environments [47]. Whole-transcriptome analysis revealed 178 differentially expressed protein-coding transcripts and several differentially expressed non-coding RNAs (lncRNAs, miRNAs, and circRNAs) between yak and cattle heart tissues. These RNAs are significantly enriched in hypoxia-related pathways, such as *VEGF* signaling. Additionally, proteins related to cardiovascular development (e.g., *ASB4*, *TSP4*) and mitochondrial function (e.g., *ACAD8*, *ALDH2*) show significantly upregulated expression under high-altitude hypoxic conditions, contributing to the maintenance of cardiac function and energy metabolism. Furthermore, genes related to energy metabolism, hypoxia adaptation, and immune regulation are significantly enriched in yak cardiac, skeletal muscle, and liver tissues under high-altitude conditions [48,49,50]. Currently, multiple copy number variations (CNVs) identified in yak research have been found to overlap with the QTLs associated with important economic traits in cattle. These affected traits include reproduction, production, meat and carcass quality, milk yield, exterior characteristics, and health, which help in pinpointing relevant functional gene loci [51]. Multiple transcriptomic and single-cell sequencing studies have collectively revealed that the molecular mechanisms underlying yak adaptation to hypoxic environments at high altitudes are the result of multi-gene regulation, demonstrating the complex regulatory system of yak adaptation to plateau environments.

To address the issue of male infertility in yak–cattle hybrids, numerous studies have focused on identifying differentially expressed genes between yaks and hybrids, utilizing transcriptomic data to uncover potential underlying mechanisms. In 2023, researchers from Sichuan Agricultural University conducted RNA sequencing (RNA-seq) and RT-qPCR validation to compare the testicular Sertoli cells of yaks and hybrids. Their findings revealed an abnormal expression of genes associated with protein activation, cellular function, and membrane organelle composition in hybrid Sertoli cells. A total of 6592 DEGs were identified, among which *Claudin-11* exhibited significantly higher expression in yak Sertoli cells compared to the hybrids. Since *Claudin-11* is closely related to spermatogenesis, this discovery provides key insights into the underlying causes of male infertility in hybrids [52]. Additionally, epigenetic studies focusing on DNA methylation and histone modifications offer further insights into how environmental factors influence gene expression, particularly in the context of hypoxia and extreme cold stress in high-altitude environments [53].To elucidate these mechanisms and validate their specific functions, it is essential to integrate multi-omics, single-cell, and temporal analyses, combined with technologies such as CRISPR, ATAC-seq, and ChIP-seq. This approach will facilitate a deeper understanding of the molecular basis of hypoxia adaptation in yak and its potential applications.

### 4.3. Gene Chip Genotyping

Gene chip technology is a rapid and accurate high-throughput sequencing method that can identify a specific number of genetic markers, especially SNPs. With the development of requirements in yak GS, Li et al. developed a genome-wide SNP array for yaks that can analyze the genetic diversity of the yak population and optimize trait selection in high-altitude environments, particularly for improving key traits such as adaptability and resistance [54]. In 2022, the Qinghai Provincial Academy of Animal Husbandry and Veterinary Science, Southwest Minzu University, and Beijing Compson Agricultural Technology Co., Ltd. (Beijing, China). jointly developed the 30K yak chip “Qingxin-1”. This chip is based on resequencing data from 80 yak samples, integrating resequencing results from breeding populations in Yushu, Huanghu, and Maiwa, as well as previously identified associated loci. The chip covers 30K functional loci related to various economic traits [55]. Yak breeding research still largely relies on bovine SNP chips and reference genomes due to limited yak-specific molecular resources. Current gaps include high-density chip design, genome assembly accuracy, and functional annotation. Bovine chips like Illumina BovineHD 770K remain cost-effective and scalable for large populations. Although a 600K yak SNP chip has been developed for trait-associated marker screening, its genome coverage and cross-population validation remain limited. Tools such as GATK, PLINK, BWA, and STAR are fully applicable to yak data, aiding genetic analysis. Advancing yak genomics and transcriptomics, along with improved reference genomes and SNP arrays, will be key to boosting molecular breeding efficiency.

## 5. Prediction Models and Heritability Estimation

### 5.1. Prediction Models

Prediction models in yak GS can generally be divided into four main types: linear models [56], Bayesian models [57], machine learning [58], and deep learning [59]. Among the linear models, gBLUP is widely applied. It assumes that all SNPs contribute equally to trait variations, which makes it suitable for large-scale yak datasets with polygenic traits. The gBLUP model has been used in some studies to predict production traits in yaks, such as weight and milk quality, with its accuracy often being higher than traditional phenotype-based selection. When applied to predict the birth weight in yaks, the accuracy can reach 0.5–0.6, while the accuracy for weaning weight is typically between 0.4 and 0.5 [60].

In Bayesian models, BayesA and BayesB are commonly used, with different prior distributions to estimate the SNP effects. BayesB allows for different variances of effects and is suitable for data with varying effect sizes [57]. In practice, the accuracy of the BayesB model may be comparable to gBLUP, but sometimes it provides higher accuracy when dealing with a small number of markers with large genetic effects [61]. The accuracy of these models is influenced by various factors, including population size, marker density, environmental effects, and the genetic structure of traits. The accuracy of the model must be assessed and validated for specific traits and populations. This involves evaluating performance using methods like cross-validation, posterior predictive checks, Bayesian Information Criterion (BIC), and predictive accuracy metrics to ultimately identify the optimal model for achieving prediction goals.

In practice, constructing a reliable reference population is hindered by the remote and fragmented distribution of yak herds, which complicates sample collection and standardization. Long-term, high-quality phenotypic data are scarce, largely due to limited tracking infrastructure and inconsistent recording by pastoralists. Moreover, systematic pedigree records are almost nonexistent, making genetic relationship estimation less accurate. These constraints limit the power of genomic prediction models and hinder the establishment of robust breeding programs tailored to the yak populations.

### 5.2. Heritability Estimation

Heritability indicates the proportion of phenotypic variation caused by genetic factors. The estimation of heritability could be influenced by factors such as the sample size, data quality, environmental control, and genetic models. Heritability estimation is an important aspect of genetic research, as it helps us understand the contribution of genetic and environmental factors to phenotypic variation [62].

Compared to beef cattle and dairy cattle, heritability estimates for the same traits in yaks tend to be lower. This is because yaks grow in complex and harsh environments. Therefore, there are more gene–environment interaction effects in the phenotypic variance components. This also results in a lower prediction accuracy for yak genomic selection for the same traits. If a model can detect gene–environment interaction effects, it would significantly improve prediction accuracy [63].

## 6. Application of Molecular and Genomic Technology in Yak Breeding

In the last several decades, yaks were selected by the traditional and molecular methods for growth and development, meat performance, body shape and appearance, and other breeding directions. These genetic traits exemplify both the evolutionary resilience of yaks in extreme environments and their significance in guiding current genetic improvement programs. From a conservation perspective, retaining these traits is crucial for preserving the genetic uniqueness of the yak as a plateau species. In breeding, such traits serve as selection markers, allowing for more targeted and sustainable genetic improvement. As climate and production pressures intensify, identifying and maintaining these adaptive features will be vital for the dual goals of biodiversity conservation and commercial trait enhancement in yak populations.

Marker-assisted selection is a genetic improvement strategy based on molecular markers. By identifying and selecting molecular markers closely associated with target traits, MAS enables the early, efficient, and precise selection of individuals, thereby accelerating the breeding process [64]. The Lanzhou Institute of Animal Husbandry and Veterinary Medicine, Chinese Academy of Agricultural Sciences, summarized the known genes and utilized MAS to carry out the breeding of hornless yaks. They selected individuals with excellent performance and hornless phenotypes from the Qinghai Plateau yak population as parental stock. A strategy combining test crossing and controlled inbreeding was adopted to establish a breeding core group.

Four mutations associated with the polled phenotype have been identified in different cattle breeds, most of which were located on chromosome 1. Among eight sires of Alpine and Scottish origin (four polled and four horned), a single candidate mutation was identified—a complex 202 bp insertion-deletion event—that showed complete association with the polled phenotype in various European cattle breeds, except for Holstein-Friesian [65]. The 80 kb duplication, which represents the most likely causal mutation for the polled phenotype in Friesian cattle, is the sole remaining variant within the shortened Friesian haplotype [66]. A 110 kb fragment duplication mutation was identified in the genome-wide association study of polled Nellore cattle [67]. Furthermore, Medugorac et al. inferred the genomes of 76 Mongolian cattle through high-density genotyping and whole-genome sequencing analysis, identifying 39 bovine haplotypes. They further refined the mapping of the polled locus to a 121 kb region (1,889,854–2,010,574 bp) and identified two key variant sites, *P1ID* and *P219ID* [68]. Studies have shown that the *FOXL2* gene is associated with horn development in ruminants. Mutations in *FOXL2* may affect horn growth, leading to the polled phenotype [14]. The identification of these mutations provides a crucial molecular foundation for further elucidating the genetic mechanism underlying the polled trait in yaks. Additionally, Jiang Yu has contributed to the theoretical framework for breeding polled cattle and sheep and has identified *OLIG1* and *RXFP2* as key target genes for the production of polled cattle and sheep through gene editing [69].

Molecular markers were used to screen the breeding population, enabling the accurate identification of individuals carrying the hornless gene for continuous generation selection. Through genotyping of the *P1ID* and *P219ID* loci, early selection of horn traits in yaks was achieved, leading to the successful development of a new hornless yak breed—Ashidan Yak. Building on this foundation, Yan Ping’s laboratory further identified key molecular markers associated with the breeding of Ashidan yaks, providing important genetic evidence for subsequent breed improvement. The research team monitored the phenotypes of individuals at different growth stages and, combined with copy number variation (CNV) analysis, found that the HPGDS gene was significantly correlated with body weight and body length. Individuals with increased copy numbers showed a marked growth advantage, indicating that this locus could serve as a candidate marker for marker-assisted selection [70]. Further analysis of the CNV of the *SOX6* gene revealed its significant correlation with chest circumference and shoulder height in Ashidan yaks. This suggests that *SOX6*-CNV could be a potential molecular marker for yak breeding [71]. Additionally, the team explored the impact of single-nucleotide polymorphisms in the *TXK* and *PLCE1* genes on the growth traits of Ashidan yaks. Using the cGPS liquid chip, they genotyped 232 Ashidan yaks and analyzed the correlation between two SNP loci (*TXK*: g.55,999,531C>T and *PLCE1*: g.342,350T>G) and various growth traits, including body weight, height, body length, and chest circumference, at different growth stages. The results showed that both were significantly correlated with growth traits [72]. The successful breeding of Ashidan yaks enriches yak genetic resources, improves farming efficiency, and advances the yak industry. It also highlights the key role of marker-assisted selection in yak breeding, serving as a model for the efficient development of new breeds with distinct economic traits. While MAS has proven useful in yak breeding, it fails to account for most quantitative traits governed by polygenes with minor effects.

The prediction accuracy and heritability analysis for different traits across yak populations indicated GS predictability and suitability at various growth stages, including Ashidan yaks at 6 and 12 months, domesticated yaks at 18 weeks, and Maiwa yaks between 2 to 6 years. The table also compares genomic selection models such as gBLUP, MtGBLUP, BayesA, BayesB, BayesCπ, and Bayes Lasso, with heritability (h^2^) listed in the final column (Table 1). Domesticated yaks at 18 weeks consistently show higher prediction accuracy and heritability for traits like WH^18W^ (Withers Height at 18 weeks), suggesting stronger genetic influence and greater potential for selection, while Ashidan yaks generally exhibit lower accuracy, particularly in the early growth stages such as BL^12W^ (body length at 12 weeks). Maiwa yaks display variable prediction accuracy across traits and ages, with the heritability values reflecting moderate to high genetic contributions in some cases, such as CG^2Y^ (chest girth at 2 years). Overall, heritability ranges from 0.07 to 0.59, underscoring the diverse genetic basis of growth traits across populations and growth stages [73,74]. Although some models perform well within specific yak populations, their cross-population generalization remains limited. Complex models have higher predictive potential with big data, but risk overfitting when data are sparse, making simpler models like gBLUP more stable. Yak-specific models that integrate genetic and ecological features, supported by large-scale data, may improve generalizability and breeding value.

The specific prediction model to be used is determined based on factors such as the scale of yak data, the complexity of target traits, computational resources, distribution of genetic effects, and heritability. For small-scale data, models with lower computational requirements such as gBLUP or random forest [75] may be preferred. For large-scale data, Bayesian models or deep learning models may be selected to improve prediction accuracy [76].

## 7. Conclusions

In this comprehensive review, we summarize the yak breeding pipeline, from phenotypic measurement and reference population building to genomic analysis and the practical use of GS and MAS. Using MAS and GS, key genes related to growth, reproduction, and high-altitude adaptation (e.g., *EPAS1*, *ADAMTS9*, and *PLAG1*) have been identified and selected. These efforts have contributed to notable progress, such as the development of polled yak lines. Despite advances in yak GS, several critical challenges remain, including the complex interaction between genetics and the environment, the need for specialized phenotype–genotype databases, and the computational demands of large datasets. Future research should focus on integrating environmental factors into GS models, improving database systems, and developing more efficient computational tools. In addition, future strategies must carefully balance the preservation of genetic diversity with the pursuit of trait specialization to ensure sustainable genetic improvement. These areas offer both challenges and opportunities for advancing yak GS.

## Figures and Tables

**Figure 1 cimb-47-00350-f001:**
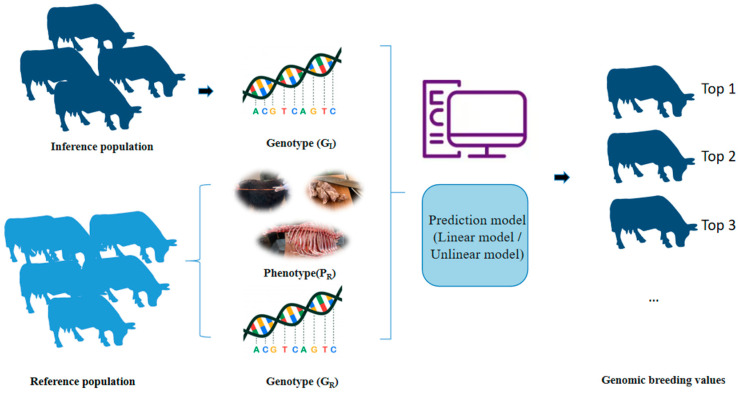
Schematic representation of genomic selection in yak breeding.

**Figure 2 cimb-47-00350-f002:**
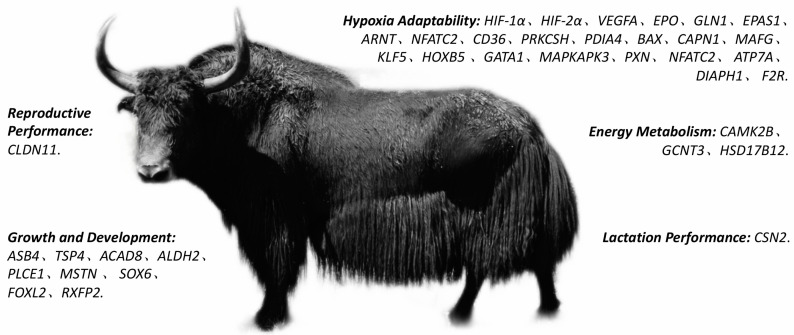
The expression of genes related to different physiological functions in yaks.

**Table 1 cimb-47-00350-t001:** Prediction accuracy and heritability of different traits in yak genomic selection.

Species	Trait	gBLUP	MtGBLUP	BayesA	BayesB	BayesCπ	Bayes Lasso	Heritability (h^2^)
	BL^6w^	0.265	NA	0.274	0.317	NA	0.293	0.560
	BW^6w^	0.277	NA	0.289	0.328	NA	0.227	0.350
	WH^6w^	0.391	NA	0.338	0.355	NA	0.379	0.570
	CG^6w^	0.234	NA	0.322	0.331	NA	0.287	0.390
Ashidan yak	BL^12w^	0.166	NA	0.166	0.155	NA	0.163	0.070
	BW^12w^	0.198	NA	0.192	0.212	NA	0.329	0.240
	WH^12w^	0.239	NA	0.168	0.177	NA	0.281	0.220
	CG^12w^	0.220	NA	0.220	0.326	NA	0.220	0.250
	BL^18w^	0.212	NA	NA	0.225	0.237	NA	0.366
	BW^18w^	0.246	NA	NA	0.247	0.264	NA	0.508
	WH^18w^	0.185	NA	NA	0.191	0.196	NA	0.589
Domesticated yak	CG^18w^	0.043	NA	NA	0.044	0.046	NA	0.345
	LYM^18w^	0.281	NA	NA	0.297	0.319	NA	0.464
	OTHR^18w^	0.197	NA	NA	0.197	0.205	NA	0.461
	PDW^18w^	0.228	NA	NA	0.238	0.246	NA	0.533
	CG^2Y^	0.183	0.496	NA	NA	NA	NA	NA
Maiwa yak	CG^3Y^	0.102	0.444	NA	NA	NA	NA	NA
	CG^4Y^	0.067	0.531	NA	NA	NA	NA	NA
	CG^5Y^	0.018	0.414	NA	NA	NA	NA	NA
	CG^6Y^	−0.065	0.450	NA	NA	NA	NA	NA

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
