# Peer review of "Decoding Quantitative Traits in Yaks: Genomic Insights for Improved Breeding Strategies"

_cimb, 2025, doi:10.3390/cimb47050350_

Round 1
Reviewer 1 Report
Comments and Suggestions for Authors
Thank you for this informative and well-structured review. The article provides a solid overview of current techniques, which are all essential for promoting genetic progress in yak populations.
To further enhance the value of the review, I would suggest expanding on a couple of important areas that currently feel underexplored. One key aspect is the lack of discussion around the unique genetic features of yaks compared to other cattle species (adaptations to high-altitude environments, enhanced oxygen utilization, cold resistance, and specific metabolic traits). This is only briefly mentioned and should be expanded. A clearer explanation of these genetic differences would help emphasize the significance of yak breeding efforts and the importance of preserving their genetic diversity.
Another area that could be further developed is the issue of hybrid male infertility in yak × cattle crosses. Since this is a major biological and practical limitation in crossbreeding strategies, it would be helpful to include more detail about the genetic or chromosomal basis of this sterility, if known.
Finally, while the review touches on many technical aspects, it raises some broader questions that might be worth addressing. For instance, what are the current challenges in developing large, genetically diverse reference populations for yak genomic studies? How well do genomic prediction models transfer across different yak populations? Are there yak-specific genomic tools being developed, or is the field still heavily reliant on cattle-based resources? Additionally, are there any logistical, economic, or ethical considerations that impact the implementation of advanced breeding programs in plateau regions?
Author Response
Comments 1: Thank you for this informative and well-structured review. The article provides a solid overview of current techniques, which are all essential for promoting genetic progress in yak populations.
Response : We appreciate your valuable time and expertise. We truly appreciate your efforts in helping us improve our manuscript. Our detailed responses are listed below.
To further enhance the value of the review, I would suggest expanding on a couple of important areas that currently feel underexplored. One key aspect is the lack of discussion around the unique genetic features of yaks compared to other cattle species (adaptations to high-altitude environments, enhanced oxygen utilization, cold resistance, and specific metabolic traits). This is only briefly mentioned and should be expanded. A clearer explanation of these genetic differences would help emphasize the significance of yak breeding efforts and the importance of preserving their genetic diversity.
Response 1: Thanks for your comments. In the revised manuscript, we have expanded the discussion on the unique genetic characteristics of yaks in comparison to other cattle species. Specifically, we have included more comprehensive descriptions of their adaptations to high-altitude environments, enhanced oxygen utilization, cold resistance, and distinctive metabolic traits. These additions are intended to better emphasize the biological significance of yak breeding efforts and underscore the importance of conserving their genetic diversity.
(Line:191-207ï¼›279-285)
Comments 2: Another area that could be further developed is the issue of hybrid male infertility in yak × cattle crosses. Since this is a major biological and practical limitation in crossbreeding strategies, it would be helpful to include more detail about the genetic or chromosomal basis of this sterility, if known.
Finally, while the review touches on many technical aspects, it raises some broader questions that might be worth addressing. For instance, what are the current challenges in developing large, genetically diverse reference populations for yak genomic studies? How well do genomic prediction models transfer across different yak populations? Are there yak-specific genomic tools being developed, or is the field still heavily reliant on cattle-based resources? Additionally, are there any logistical, economic, or ethical considerations that impact the implementation of advanced breeding programs in plateau regions?
Response 2: Thank you for your comments. In response, We have added a discussion on the issue of male infertility in yak × cattle hybrids, including the current research focus and potential chromosomal or genetic causes. Furthermore, we have expanded the manuscript to include key challenges in establishing genetically diverse reference populations for yaks, the transferability of genomic prediction models across different yak populations, the current status of yak-specific genomic tool development, and the logistical, economic, and ethical considerations involved in implementing advanced breeding programs in high-altitude regions.
Reviewer 2 Report
Comments and Suggestions for Authors
The title document does not seem like the review and does not represent what the author wanted to describe. A list of genes/qtls in yaks is missing,
Many definitions are general in genetics, so explaining them in the document does not improve the content of i
Author Response
Comments 1: The title document does not seem like the review and does not represent what the author wanted to describe.
Response 1: Thank you for your suggestion. We agree with your comments. The title has been changed to better reflect the core focus of the study:" Decoding Quantitative Traits in Yaks: Genomic Insights for Improved Breeding Strategies".
(Line:2-3)
Comments 2: A list of genes/qtls in yaks is missing, many definitions are general in genetics, so explaining them in the document does not improve the content of i.
Response 2: Thank you for your suggestion.We have also removed redundant general definitions that are commonly understood in genetics and added a dedicated section summarizing key genes and QTLs identified in previous yak studies, which are relevant to economically important traits and genomic selection efforts. We believe these revisions enhance the specificity and scientific relevance of the review.
(Line:302-306)
Reviewer 3 Report
Comments and Suggestions for Authors
Even the issue of genomic selection and genomic prediction is still interesting, I had problem to follow the manuscript as it is organized in unusual way - even it is submitted as review.
I am missing more informtion obut yak breeding sector, importance of yak farming or yak basic population data to be able to evaluate contribution to the field.
As the high altitude animal based protein availabiity is limited, I can imagine that farmed yak could serve perfectly for that purpose.
There are big differences in the organization of particular paragraphs and the proportionality each to other. It is hard to follow and separate methodological, technical and results part.
In case of results it is just repetiton (copy) of original outcomes without any scientific discussion over them.
However, this review has potential to provide copressed information on one place about GS and GP in yak but in current state is is more schollarly work than scientific review and I would recommend to spend some more time and re-organize the work in usual form (IMRAD).
Regarding the his review has potential to provide copressed information on one place about GS and GP in yak but in current state is is more schollarly work than scientific review and I would recommend to spend some more time and re-organize the work in usual form
Author Response
Comments 1: Even the issue of genomic selection and genomic prediction is still interesting, I had problem to follow the manuscript as it is organized in unusual way - even it is submitted as review... without any scientific discussion over them.
Response 1: Thank you for your suggestions. We have followed your comments to revise sentence.The manuscript has been re-organized following the IMRAD structure to improve logical flow and readability.
(Throughout the manuscript)
Comments 2: I am missing more informtion obut yak breeding sector, importance of yak farming or yak basic population data to be able to evaluate contribution to the field.
Response 2: Thank you for your comments. We have added background information on the yak breeding sector, including the significance of yak farming in high-altitude regions, population size, and current breeding practices. This addition provides essential context for evaluating the contribution of genomic technologies in yak.
(Line:32-43)
Comments 3: As the high altitude animal based protein availabiity is limited, I can imagine that farmed yak could serve perfectly for that purpose.
Response 3: Thank you for your comments. we have emphasized the role of yak as a key source of animal protein in high-altitude regions where alternatives are scarce.
(Line:36-39)
Comments 4: There are big differences in the organization of particular paragraphs and the proportionality each to other. It is hard to follow and separate methodological, technical and results part.
Response 4: We acknowledge the issue with the previous structure. This review is structured around six key components: phenotypic assessment, sample collection, reference population design, genotyping, predictive modeling, and heritability estimation. By this way,we can systematically evaluate critical components of genomic breeding pipelines. In addition, we have replaced the original "Outlook & Future" section with a "Conclusion" section, which summarizes the main content, discusses key points, and provides suggestions for future research.
(Line:480-494)
Comments 5: In case of results it is just repetiton (copy) of original outcomes without any scientific discussion over them.
Response 5: We agree with your viewpoint. Instead of repeating original findings, we now provide scientific interpretation and discuss their implications for yak breeding, highlighting strengths and gaps in the current literature.
(Throughout the manuscript)
Comments 6: However, this review has potential to provide copressed information on one place about GS and GP in yak but in current state is is more schollarly work than scientific review and I would recommend to spend some more time and re-organize the work in usual form (IMRAD). Regarding the his review has potential to provide copressed information on one place about GS and GP in yak but in current state is is more schollarly work than scientific review and I would recommend to spend some more time and re-organize the work in usual form
Response 6:Thank you for your comments. We have addressed this by refining the structure, improving scientific discussion, and enhancing the depth of analysis throughout the manuscript. These changes elevate the manuscript from a scholarly summary to a formal scientific review.
(Throughout the manuscript)
Round 2
Reviewer 3 Report
Comments and Suggestions for Authors
Thank you very much for providing revised version, however I have to disagree with your responses as the paper was not reorganized (Throughout the manuscript) as presented:
1) Reorganize chapters to be in logical order from Introduction, to Methodological approaches, Results and Discussion, whereas the last mentioned should be major part of the review.
2) Author mention only one way of biological samples collection? Are there any alternatives in case of wild yak's? Or only slaughtered animals are in the studies?
3) Discussion is missing - can you please make any statements over methods and models used in GS and GP? What about the accuracy of models presented any statement? Is it good as it is? Really?
You added text describing data and hading of the tables but no explanation or discussion.
I am sorry but in present stage, I dont have more possitive review for you.
Comments on the Quality of English Language
English level is sufficient.
Author Response
Comments 1: Reorganize chapters to be in logical order from Introduction, to Methodological approaches, Results and Discussion, whereas the last mentioned should be major part of the review.
Response 1: We thank the reviewer for the thoughtful comment. However, as this manuscript is designed as a narrative review rather than a systematic review, it follows a thematic and conceptual organization rather than a strict IMRaD structure. We have clarified this in the Introduction section to avoid potential confusion. We respectfully suggest maintaining the current structure, which we believe better suits the purpose and nature of a narrative review. We also noticed that several review papers were similar format to ours[1–3].
Comments 2: Author mention only one way of biological samples collection? Are there any alternatives in case of wild yaks? Or only slaughtered animals are in the studies?
Response 2: We appreciate the reviewer’s valuable comment. However, we would like to clarify that our review focuses specifically on breeding practices in domestic yaks, and thus does not cover wild yak populations. We agree that multiple types of biological samples can be relevant in this field. In response to the comment, we have now supplemented the manuscript to include additional common sampling approaches such as blood, hair follicles, feces, and mucosal swabs, which are frequently used in genomic and reproductive studies of domestic yaks.
(Line:216-226)
Comments 3: Discussion is missing - can you please make any statements over methods and models used in GS and GP? What about the accuracy of models presented any statement? Is it good as it is? Really?
Response 3: We thank the reviewer for the insightful comment. In response, we have added a new Discussion section named Application of Molecular and Genomic Technology in Yak Breeding to the manuscript, in which we summarize and critically analyze the methods and models used in genomic selection (GS) and genomic prediction (GP) for yak breeding. We also discuss the accuracy of commonly used models, their current performance, and potential areas for future improvement.
(Line:372-378ï¼›391-483)
Comments 4: You added text describing data and hading of the tables but no explanation or discussion.
Response 4: We have followed your comments to supply explanation and discussion of table and figures in the manuscript. We also moved table and figures to right position and marked it citation in the manuscript.
(Line:112-119ï¼›262-265ï¼›475-476)
Comments 5: I am sorry but in present stage, I dont have more possitive review for you.
Response 5: We thank your contribution in guiding us to revise our manuscript. We have adjusted the structure of this manuscript and added a Discussion section. Hope that it meets your requirements.
(Line:391-483)
Reference:
- Moon D. Targeting SHP2 with Natural Products : Exploring Saponin-Based Allosteric Inhibitors and Their Therapeutic Potential. 2025.
- Tataranu LG. Harnessing Exosomes : A Brief Overview of Nature ’ s Nanocarriers and a Glimpse into Their Implications in Pituitary Neuroendocrine Tumors ( PitNETs ). 2025;1–16.
- Qadir BH, Abdulghafor MA, Mahmood MK, Zardawi FM, Fatih MT, Kurda HA, et al. Applications of Growth Factors in Implant Dentistry. 2025;1–19.
Round 3
Reviewer 3 Report
Comments and Suggestions for Authors
I have no furhter comments or questions.